# Genome-Wide Identification, Comparison, and Expression Analysis of Transcription Factors in Ascidian *Styela clava*

**DOI:** 10.3390/ijms22094317

**Published:** 2021-04-21

**Authors:** Jin Zhang, Jiankai Wei, Haiyan Yu, Bo Dong

**Affiliations:** 1Sars-Fang Centre, MoE Key Laboratory of Marine Genetics and Breeding, College of Marine Life Sciences, Ocean University of China, Qingdao 266003, China; zhangjincoolcool@163.com (J.Z.); weijiankai@ouc.edu.cn (J.W.); 2Laboratory for Marine Biology and Biotechnology, Qingdao National Laboratory for Marine Science and Technology, Qingdao 266237, China; 3Institute of Evolution and Marine Biodiversity, Ocean University of China, Qingdao 266003, China

**Keywords:** *Styela clava*, transcription factor, zinc-finger protein, *hox*, *forkhead box*

## Abstract

Tunicates include diverse species, as they are model animals for evolutionary developmental biology study. The embryonic development of tunicates is known to be extensively regulated by transcription factors (TFs). *Styela clava*, the globally distributed invasive tunicate, exhibits a strong capacity for environmental adaptation. However, the TFs were not systematically identified and analyzed. In this study, we reported 553 TFs categorized into 60 families from *S. clava*, based on the whole genome data. Comparison of TFs analysis among the tunicate species revealed that the gene number in the zinc finger superfamily displayed the most significant discrepancy, indicating this family was under the highly evolutionary selection and might be related to species differentiation and environmental adaptation. The greatest number of TFs was discovered in the Cys2His2-type zinc finger protein (zf-C2H2) family in *S. clava*. From the point of temporal view, more than half the TFs were expressed at the early embryonic stage. The expression correlation analysis revealed the existence of a transition for TFs expression from early embryogenesis to the later larval development in *S. clava*. Eight *Hox* genes were identified to be located on one chromosome, exhibiting different arrangement and expression patterns, compared to *Ciona robusta* (*C. intestinalis* type A). In addition, a total of 23 *forkhead box* (*fox*) genes were identified in *S. clava*, and their expression profiles referred to their potential roles in neurodevelopment and sensory organ development. Our data, thus, provides crucial clues to the potential functions of TFs in development and environmental adaptation in the leathery sea squirt.

## 1. Introduction

Transcription factors (TFs) specifically recognize the DNA sequence through DNA-binding domains (DBDs) on itself, controlling transcription, and performing the first step in decoding the DNA sequence [1]. All TFs have at least one DBD, through which they attach to a specific sequence of DNA fragment adjacent to the genes, and then the transcriptions of these genes are either activated or repressed [2]. DBD is a crucial standard for the identification and classification of TFs. The DBD, regulatory regions, and biological functions of TFs are largely conserved across metazoans, although the number and diversity of TFs in different organism are variable [2,3,4]. TFs play extensively and essentially regulatory functions in diversely biological processes, such as cell differentiation [5], organ development [6], inflammatory response [7], and body-axis building [8]. In metazoans, TFs were identified and classified into 78 TF families in the REGULATOR database [9]. Genome-wide TFs databases are also available, such as the Animal Transcription Factor DataBase (Animal TFDB) [10]. In these databases, TFs of the representative species, such as *Homo sapiens* and *Drosophila melanogaster* are identified and classified.

Tunicates occupy a crucially phylogenetic status between invertebrate and vertebrate, which are closely related to the vertebrates in evolution. The characteristic development process makes them a great study model for developmental biology. In *Ciona robusta* (*C. intestinalis* type A), TFs were identified at the genome level [11,12,13,14,15,16]. Most *Ciona* TFs are expressed maternally and were demonstrated to regulate early embryonic development. For example, *brachyury*, belonging to the T-box TF family, regulates the notochord cell specification and morphogenesis, through the downstream genes mediated by cis-regulatory modules [17,18,19,20], such as the *nuclear factor of activated T cells 5* (*nfat5*), *T-box2/3* (*Tbx2/3*), and *signal transducer and activator of transcription 5* (*stat5*) [21,22]. Whereas, *FoxD* activates the expression of the *zinc finger of the cerebellum-like* (*ZicL*) part, which binds and promotes the expression of *brachyury* [23,24,25].

The various TFs in different animals were evolutionarily related to their regulatory ways. For example, *hox* genes were conserved throughout vertebrates [26,27], but in tunicates, the number and arrangement of *hox* genes were quite different, indicating the different roles of *hox* genes in tunicates [26,27].

*Styela clava* is a globally distributed ascidian species that shows a strong environmental adaptation [28,29,30]. In this study, we performed genome-wide identification and analysis for TFs in *S. clava*. The expanded and contracted TF families were also identified through comparison with other species. Furthermore, the expression profiles of TFs in *S. clava* were also analyzed. These results provided insights into the understanding of the regulatory mechanisms of TFs in embryogenesis and environmental adaptation in tunicates, and the evolution of the TF families.

## 2. Results

### 2.1. Identification of TFs in the S. clava Genome

A total of 17,428 protein-coding genes were identified, 15,734 of which were annotated in the *S. clava* genome [28]. Based on our previous study, we further identified 553 TF genes, which were distributed in 60 TF families in the *S. clava* genome, according to the domain annotation of proteins and the types of DBD. In these 60 TF families, the Cys2His2-type zinc finger protein (zf-C2H2) family had the largest number of TFs, with 154 genes (27.85% of total TFs) (Table 1). The Homeodomain family and the basic helix–loop–helix (bHLH) family were the second and third largest TF family, with 73 (13.20%) and 41 (7.41%) TFs, respectively (Table 1). There were 16 orphan TFs, belonging to the TF family with only one member including the ALL1-fused gene from chromosome 4 (AF-4), the CCAAT-binding transcription factor subunit B_nuclear transcription factor Y subunit alpha (CBFB_NFYA), and interferon regulatory factors-3 (IRF-3), etc. (Table 1).

### 2.2. Comparison Analysis of S. clava TFs

To compare the TFs in *S. clava* with other species, the same approach was utilized to identify TFs in the species genomes. Three species, including human (*H. sapiens*), cephalochordate (the lancelet, *Branchiostoma floridae*), nematode (*Caenorhabditis elegans*), and five tunicates, including *C. robusta*, *Molgula oculata*, *Oikopleura dioica*, *Botrylloides leachii*, and *Botryllus schlosseri* were selected for comparison analysis. The results showed that there were 51 families shared by *S. clava*, *H. sapiens*, *B. floridae*, and *C. elegans* (Figure 1A, Appendix A). There were also 51 families shared among the five ascidian species (Figure 1B, Appendix A). There were 44 gene families shared between these two groups. Seven gene families including Transcriptional Coactivator p15 (PC4), Cysteine/serine-rich nuclear protein N-terminus (CSRNP_N), DM, CBFB_NFYA, Helix-turn-helix (HTH_psq), Heat shock factor type DNA binding (HSF_DNA-binding), and the MIZ-type zinc finger (zf-MIZ) were only shared among the species in Figure 1A, while the other seven gene families including RHD_DNA_binding, BEAF and DREF-type zinc finger protein (zf-BED), AF-4, Homeo-prospero domain (HPD), IRF-3, IRF, and NLS-bindingm and DNA-binding, and the dimerization domains of Nrf1 (Nrf1_DNA-binding) were only shared among the species in Figure 1B (Appendix A).

To explore the expansion and contraction of TFs families, we compared the TFs families in tunicates with other species. Nine human TF families could not be found in tunicates and *C. elegans* (Table 1). Among these nine families, two were found in *B. floridae* (Table 1). The thanatos-associated proteins (THAP) family was increased in Stolidobranchia ascidians, including *S. clava*, *M. oculata*, *B. leachii*, and *B. schlosseri*, compared to *H. sapiens* (Table 1). Compared to *B. floridae* and *C. elegans*, general transcription factor IIi (GTF2I), Signal Transducer, and the Activator of Transcription binding (STAT_binding) families were only found in tunicates (Figure 1A, Table 1 and Appendix A).

In the *S. clava* genome, six TF families had the largest number of TFs among the six tunicates, including the bHLH, the Cold shock domain (CSD), the CTF/NF-I family transcription modulation region (CTF_NFI), NHR1 homology to TAF (TAFH), zf-BED, and Sp100, AIRE-1, NucP41/75, and DEAF-1 (SAND) families (Table 1). Two TF families in the *S. clava* genome had the least number of TFs among the six tunicates, including the forkhead box (fox) and Leucine-Rich Repeat in the Flightless-Interaction Protein (LRRFIP) families (Table 1). The number of TFs in *S. clava* were more than that in *C. robusta*, *O. dioica*, and *B. leachii* (Table 1). The expansion mainly concentrated on the zf-C2H2 family, in which 59, 82, and 44 more TFs were identified in *S. clava* than that in *C. robusta*, *O. dioica*, and *B. leachii*, respectively (Table 1). Overall, comparison analysis revealed that the gene number of the zf-C2H2 family showed the most significant change among tunicates, indicating that the zinc finger family genes were under adaptative changes.

### 2.3. Expression of S. clava TFs at Different Developmental Stages

To explore the expression profiles of TFs during the development of *S. clava*, we acquired the fragments Per Kilobase of gene per million mapped reads (FPKM) value of each TF in 2-cell–8-cell embryos (2–8 cells), gastrula embryos (gast), neurula embryos (neu), tailbud-stage embryos (tb), hatched swimming larvae (hsl), tail-regressed larvae (trl), and metamorphic juveniles (mj) by RNA-sequencing [28]. These data were further validated through quantitative real-time PCR. The results showed that the relative expression levels of the randomly selected genes were coincident with the results of FPKM values (Appendix A). To analyze the expression profiles of TFs in *S. clava* during early development, we clustered 547 TFs by the weighted correlation network analysis (WGCNA) method [31] and visualized the expression level by a heat map (Figure 2). The TF genes were clustered into nine modules, labeled with different colors, based on the results of the WGCNA analysis. The turquoise module contained the most TFs, nearly one third of TFs, which were highly expressed at the 2–8 cells stage and the gast stage (Table 2). A total of 290 TF genes (52.44% of total TF genes) were expressed (with FPKM value >10) at the 2–8 cell stage in *S. clava*, about 161 of them were classified in the turquoise module. The expression level of these genes in this module decreased from the embryo to the larvae stage. The brown module contained nearly one fifth of TFs that were mainly expressed at the tb stage (Table 2, Figure 2).

Based on the expression profiles of each module, we categorized nine modules into three groups. Group I contained TFs that were highly expressed during 2 cell stage to neu stage, including turquoise, blue, magenta, and pink modules. Group II contained TFs that were highly expressed during the tb stage, including the brown module. Group III contained TFs that were highly expressed during the hsl stage to mj stage, including the yellow, red, green, and black modules. More than half of the TFs (58.94%) were clustered into Group I, almost three times than that in Group II (18.25%) and Group III (22.81%) (Table 2). In Group I, zf-C2H2, homeodomain, and the bHLH family were the largest TFs (Table 2). In Group II, homeodomain and the bHLH families were the largest TFs (Table 2). In Group III, homeodomain, zf-C2H2, and fox families were the largest TFs (Table 2). Overall, the homeodomain TF family was dispersed into each group, while the Fox family was highly expressed during the larval development in Group III.

According to the expression correlation heat map of TFs, we found that five boxes of TFs showed significant correlation, which are indicated in box A to E (Figure 3). We found that the TFs in Group I were mainly distributed in box A, box D, and box E, TFs in Group II and Group III were mainly distributed in box C and box B, respectively (Appendix A). The expression correlation results showed that the TFs expressed in the embryogenesis and larvae development had a low correlation, indicating the discrepancy of TFs in regulating the process of embryogenesis and larvae development.

### 2.4. Hox Genes

*Hox* genes are important members of the homeobox TF family and share common development mechanisms in regulating the anteroposterior body axis [8]. In vertebrates, there are 13 paralog groups (PGs) for *Hox* genes. Through phylogenetic analysis and blast annotation, we identified eight *Hox* genes in *S. clava*, including *ScHox1*, *ScHox2*, *ScHox3*, *ScHox4*, *ScHox5*, *ScHox10*, *ScHox12*, and *ScHox13*, which were classified into eight PGs (Appendix A). The molecular phylogenetic tree provided a convincing evidence that *ScHox1*, *ScHox2*, *ScHox3*, *ScHox10*, *ScHox12*, and *ScHox13* belonged to the PG1, PG2, PG3, PG10, PG12, and PG13, respectively (Appendix A). *ScHox4* and *ScHox5* were identified by sequence alignment. The results showed that they belonged to PG4 (*e*-value = 2 × 10^−57^) and PG5 (*e*-value = 5 × 10^−52^), respectively. All *Hox* genes identified in the *S. clava* genome were distributed on one single chromosome according to the Hi-C result [28]. *Hox* genes were also distributed on one single chromosome in *Halocynthia roretzi* with similar arrangement, and on two chromosomes in the *C. robusta* genome (Figure 4A). In other tunicates mentioned in this study, the *Hox* genes were distributed on several scaffolds (Figure 4A). In *H. sapiens*, four Hox clusters existed in the genome and were distributed on four chromosomes (Figure 4A). *B. floridae* contained all 13 kinds of *Hox* genes and were distributed on one single chromosome (Figure 4A). In tunicates, there are nine *Hox* genes in *C. robusta*, *O. dioica*, *H. roretzi*, and *B. schlosseri* genome [32,33,34,35]. There are six, seven, and eight Hox genes in the *M. oculata*, *B. leachii*, and *S. clava* genome [27,35], respectively (Figure 4A).

The expression profiles of *Hox* genes in *S. clava* showed that *ScHox4* and *ScHox12* were initially expressed at the neu stage, *ScHox1*, *ScHox10*, and *ScHox13* were initially expressed at the tb stage, *ScHox3* was initially expressed at the hsl stage, and expression values of *ScHox2* and *ScHox5* were low during early development (Figure 4B). Expression of *ScHox13* were restricted at the tb stage (Figure 4B). While *ScHox12* were not expressed at the hsl stage and the trl stage, after initial expression at the neu stage (Figure 4B). Among the sub-cluster of *ScHox2*, *ScHox3*, and *ScHox4*, *ScHox4* was initially expressed at 2–8 cells stage, earlier than the expression of *ScHox3* and *ScHox2* that were expressed at the gast stage and tb stage (Figure 4B).

### 2.5. Zinc Finger Family

Zinc finger superfamily contains the greatest number of TFs in the *S. clava* genome (Table 1). The quantity variation of zinc finger TFs was the main contributor to the different number of TFs between species that we mentioned. There were 11 families in the zinc finger superfamily, including THAP, zf-BED, zf-C2H2, etc. (Table 1). Among these families, the THAP family was expanded in the tunicate genome and the zf-BED family was expanded in the *S. clava* genome, among the nine species we analyzed (Table 1).

The Zf-C2H2 family was the largest family in the *S. clava* genome, and also in the other eight species we analyzed (Table 1). According to the domain analysis, there are two kinds of zf-C2H2 proteins in the *S. clava* genome, including ZBTB and zf-C2H2. The ZBTB proteins were characterized by containing two domains, the BTB domain and zf-C2H2 domain. According to the feature of ZBTB, we identified 12 ZBTB genes in the *S. clava* genome, more than the other six tunicates analyzed (Figure 5A).

The *zf-C2H2* genes were the majority of the zf-C2H2 family. There are 132 *zf-C2H2* genes mapped on 16 chromosomes of the *S. clava* genome (Figure 5B). The Chr 5 processed the greatest number of *zf-C2H2* genes (Figure 5B). Those *zf-C2H2* genes, which were highly expressed after the neu stage (red rectangles), were located on nine of 16 chromosomes, among which Chr 3 was the most common. The wide distribution of the zf-C2H2 genes indicates the crucial regulatory roles of the zf-C2H2 domain on activating the downstream gene expression in various biological processes.

### 2.6. Forkhead Box Family

A total of 23 *Fox* genes were identified in the *S. clava* genome and they could be grouped into 15 *Fox* subclasses (Figure 6 and Appendix A). The largest *Fox* subclass was FoxI, which contained four *ScFox* genes (Figure 6). The expression profiles of the *ScFox* genes were varied during early development (Figure 6, Table 2). Five of them were highly expressed before the tb stage, eight were highly expressed during the tb stage, and ten were highly expressed during larvae development (Figure 6). In the *S. clava* genome, we could not find the homologous genes of *FoxB*, *FoxK*, *FoxL*, and *FoxS*. The expression profiles showed that the *ScFox* genes were highly expressed in several stages during early development. This expression pattern was also presented in the Yesso scallop (*Patinopecten yessoensis*) [36] and sea urchin [37]. Similar patterns presented in early development in these species indicated that *ScFox* genes are widely involved in regulating the processes of early development of embryogenesis.

## 3. Discussion

Tunicates, as the closest relatives of vertebrate, show a special rates and patterns of molecular evolution [38]. Identification and analysis of important gene families were also performed, based on the sequenced genomes in several tunicate species [35,39]. In this study, we screened and identified 553 TFs in the *S. clava* genome and revealed their potential roles in environmental adaptation and early embryonic development, through their expression profiles.

*S. clava* showed a broad tolerance of environmental conditions [28]. Compared to *B. floridae* and *C. elegans*, there are more CSD TFs and immune response-related TFs, (such as *IRF* genes) in *S. clava*. CSD is a crucial characteristic of cold shock proteins (CSP), which were upregulated under the low temperature in *S. clava* [28]. *IRF* genes play an important role in immune response in *C. savignyi* [40,41]. The expansion of these TFs might help to improve the fitness to low temperature conditions and improve their ability of immune response to adapt to new environment.

TFs highly expressed during 2–8 cells stage is an important group that we identified in this study. Through gene ontology (GO) enrichment analysis, the biology processes of maternal TFs mainly concentrate in the response/cellular response, signaling, and regulation of biological process (Appendix A). More than half of TFs involved in these processes are annotated as nuclear receptors, which regulate the activation of genes and control many biological processes, such as cell proliferation, cell cycle, and metabolism [42]. For example, thyroid hormone receptor α1 (THRα1) controls the cell proliferation through the Wnt /β-catenin pathway [43]. In *C. robusta*, the combination of GATA and Ets induced FGF9/16/20 in the neural tissue [44,45,46,47]. TFs highly expressed during 2–8 cells stage in *S. clava* might act as “responders” and “signal transmitters” to control the expression of the downstream genes and regulate early embryogenesis.

Proportion of the expressed TFs had increased from 2–8 cells stage to the tb stage (Appendix A) and was stable from the hsl stage to the mj stage. In the previous study, the different species showed very similar overall TF expression patterns, with TF expression increasing during the initial stages of development among *D. rerio*, *C. robusta*, *D. melanogaster*, and *C. elegans* [48]. We compared the TF expression between *S. clava* and *C. robusta* and found similar patterns, with TF expression increasing from 2–8 cells stage to tb stage (Appendix A). These results indicate that the most TFs might play a conserved role in early development.

*S. clava* larvae undergo metamorphosis after hatching. TFs, whose expression are significantly up-regulated after hatching might be involved into the regulation of this process. We classified those genes into two types. One is immune response TFs, which include cyclic AMP-dependent transcription factor (ATF-3), IRF1, IRF4, and B-cell lymphoma 6 (BCL6), which have strong activities in immune response regulation and maturation of the immune system [49,50,51,52]. High expression of these TFs indicates that the immune system and inflammatory reaction are involved in the metamorphosis of *S. clava*, and these immune responses also appeared in ascidian *Boltenia villosa* [53]. The others are the thyroid hormone, retinoic acid signaling TFs, and nuclear receptors, including thyroid hormone receptor alpha (THRα), vitamin D 25-hydroxylase (CYP2R1), and retinoic acid receptor alpha (RARα). Thyroid hormone and retinoic acid signaling pathways were demonstrated to be essential in the metamorphosis of *S. clava* [28,54]. Nuclear receptors are a superfamily of TFs that function in the regulation of various metabolic processes [42,55]. The metamorphosis process is regulated through interaction of hormone and nuclear receptors, which also existed in fishes [56]. Nuclear receptors, involved in the TH and RA signaling pathways, had potential in regulating metamorphosis in *S. clava*, according to the expression profiles.

Eight *ScHox* genes were identified in the *S. clava* genome and they were located in one chromosome. This was consistent with the location of *Hox* genes in *H. roretzi*, while *Hox* genes in genome of *C. robusta* were located in two chromosomes [27,33,35]. The similar location of *Hox* genes on the chromosome between *S. clava* and *H. roretzi* might be related to their close phylogenetic relationship [28]. Whole-cluster temporal collinearity (WTC) is a typical expression characteristic for *Hox* genes in vertebrate [57,58]. However, it is not applicable in tunicates. Expression profile of *Hox* genes in *S. clava* showed a subcluster temporal collinearity (STC), which was reflected in the *ScHox*2–4 cluster. STC was also presented in other invertebrate species, such as scallop *P. yessoensis* [59] and ascidian *C. robusta* [32]. In *C. robusta*, the *Hox* genes were spatially expressed in the central nervous system of larvae and the gut of the juvenile [32]. *CrHox10* and *CrHox12* play important roles in the neuronal and tail development in *C. robusta* [60]. These investigations suggest the potential roles of *Hox* genes in the development of the tail and the nervous system in *S. clava*.

The expression profiles of *Hox1*, *Hox3*, *Hox4*, *Hox10*, and *Hox12* genes in *S. clava* were similar to the expression profiles of these genes in *C. robusta* [32]. The expression of *Hox2* and *Hox5* were very low in *S. clava*. The expression profiles of *Hox13* were different between *S. clava* and *C. robusta*. *ScHox13* was expressed in the tb stage, but *CrHox13* was detectable only in the juvenile [32]. However, the number and distribution of *Hox* genes seemed not to be related to their body plan at the larval stage.

*Fox* genes are characterized by the highly conserved forkhead motif, which is known to be a “winged-helix” DNA-binding domain [61,62]. *Fox* genes play important roles in embryogenesis and metabolism [62]. A total of 22 *Fox* genes were identified in the genome of sea urchin [37]. Number of *Fox* genes in *S. clava* genome was nearly same as that of the sea urchin. Compared to the sea urchin, *FoxE*, *FoxH*, and *FoxR* subclasses in *S. clava* were found, and *FoxI* subclass was expanded, but *FoxB*, *FoxK*, and *FoxL* subclasses were not found. In *S. clava*, genes in the *FoxE* and *FoxI* subclasses were highly expressed after hatching. In vertebrates, *FoxE4*, *FoxI1*, and *FoxI2* play important roles in len, otic placode, and retina formation and development, respectively [63,64,65]. *ScFoxE* and *ScFoxI* might have similar functions in the regulation of neuro and sensory organ formation.

In conclusion, we identified 553 TFs belonging to 60 TF families in the *S. clava* genome. The expression profiles provided a possible clue for the functions of different TF genes in embryogenesis, environmental adaptation, and metamorphosis in *S. clava* (Figure 7). Our study provides gene resources and a new perspective to understand the evolution and function of TFs in the leathery sea squirt.

## 4. Materials and Methods

### 4.1. Animals and Embryos

The adults of *S. clava* were collected from Weihai City, China, and cultured in seawater at 18 °C in the laboratory. The eggs and sperm were collected separately from different individuals for fertilization at room temperature. The embryos and larvae were collected for RNA extraction at different stages. The study was approved by the Ocean University of China Institutional Animal Care and Use Committee (OUC-IACUC) prior to the initiation of the study (Approval number: “2021-0032-0012”, 15 April 2019). All experiments and relevant methods were carried out in accordance with the approved guidelines and regulations of OUC-IACUC.

### 4.2. Transcriptome Sequencing and WGCNA Analysis

The transcriptome sequencing and data were described in a previous study [28]. The co-expression gene network for 21 transcriptomic datasets was constructed using the R package WGCNA, with the parameters of softPower = 12, minimum module size = 300, cutting height = 0.99, and deepSplit = F [31]. The genes that were expressed in at least one developmental stage were used for network construction.

### 4.3. Identification and Classification of TFs

The TFs were identified and classified, based on the conserved DBDs. After obtaining all protein sequences at the genome level, we removed random sequences through CD-HIT (v4.6.8) [66] and analyzed the protein domain according to the Hidden Markov Model (HMM) profiles from the Pfam database (version 31.0) [67], applying the hmmscan program in the HMMER package(v3.1b2) [68]. 85 Pfam ID of DBDs from REGULATOR database (http://www.bioinformatics.org/regulator/page.php?act=family, accessed on 8 December 2020) and the Animal TFDB database (http://bioinfo.life.hust.edu.cn/AnimalTFDB/#!/species, accessed on 8 December 2020) were used for analysis [9,10]. The Pfam IDs were showed in Appendix A. Most genes containing DBDs were screened with the *e*-value threshold of 10^−4^ and were regarded as TFs. Some of thresholds of DBD were chosen according to the threshold in Animal TFDB 3.0 [10], including the threshold of the bHLH domain with 10^−2^, thresholds of the HMG, Homeodomian, zf-BED, zf-C2H2 domains with 10^−3^, and the thresholds of the zf-CCCH domains with 10^−20^. The TFs were classified into different families, according to the DBDs.

### 4.4. Heat Map, Phylogenetic Analysis, Domain Analysis, and Expression Correlation Analysis

We normalized the FPKM values through log10(FPKM + 1) and imported the normalized values into the Heml software (v1.0) [69]. We performed molecular phylogenetic tree analysis using the MEGA software (v7.0) [70] through the Maximum-Likelihood (ML) method and beautified the figure through iTOL (https://itol.embl.de/, accessed on 13 December 2020) online [71]. The multiple sequence alignments were conducted by Clustal W [72]. The gene expression correlation Heat map were constructed by the corrplot package and the ggcorrplot package on R studio. The correlation coefficient indicated the correlation between two genes. The correlation coefficient was greater than 0 for positive correlation (shown in red), and less than 0 for negative correlation (shown in blue). (https://github.com/taiyun/corrplot, accessed on 21 December 2020).

### 4.5. Quantitative Real-Time PCR

Total RNA was extracted through the TRIZOL method. Reverse transcription experiments were performed according to HiScript II Q RT SuperMix for QPCR (+gDNA wiper) (Vazyme R223-01, Nanjing, China). 18S rRNA was chosen as an internal standard. The primers used in experiments are shown in Appendix A. Quantitative real-time PCR was carried out by the ChamQ SYBR Color QPCR Master Mix (Vazyme Q411-01, Nanjing, China). The relative gene expression levels were calculated using the comparative Ct method with the formula 2^−∆∆Ct^ [73].

### 4.6. Gene Ontology (GO) Enrichment Analysis

The GO enrichment analysis was performed by the OmicShare tools online (https://www.omicshare.com, accessed on 23 September 2020). The target genes were maternal expressed TF genes, the background genes were all TF genes. We obtained the top 20 of GO enrichment results for further analysis and made the bubble chart.

### 4.7. Data Availability

The genome sequences of *S. clava* were deposited in NCBI, under the BioProject number PRJNA523448. The transcriptome data of *S. clava* used in this study were also deposited in the NCBI SRA database, with the accession numbers SRR8599814 to SRR8599834.

The genome resources of *C. robusta*, *H. sapiens*, and *C. elegans* were downloaded from the Ensembl (https://asia.ensembl.org/index.html, version101, accessed on 1 July 2020) [74]. The genome resources of *O. dioica* and *B. floridae* were acquired from NCBI (https://www.ncbi.nlm.nih.gov/, accessed on 1 July 2020), and the genome resources of *M. oculata*, *B. leachii*, and *B. schlosseri* were acquired from ANISEED (https://www.aniseed.cnrs.fr/, accessed on 1 July 2020) [75].

## Figures and Tables

**Figure 1 ijms-22-04317-f001:**
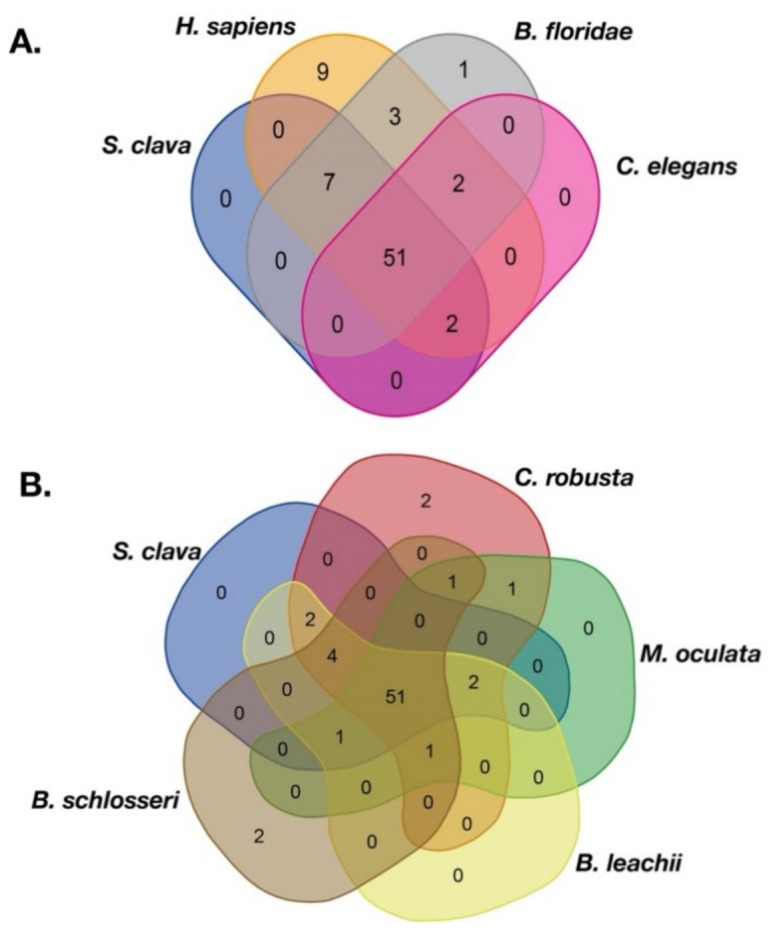
Veen of the TF families among different species. (**A**) TF families among *S. clava* (blue), *H. sapiens* (dark yellow), *B. floridae* (gary), and *C. elegans* (pink). (**B**) TF families among *S. clava* (blue), *C. robusta* (red), *M. oculata* (green), *B. leachii* (yellow), and *B. schlosseri* (brown). The transcription factor families contained in each part are listed in Appendix A.

**Figure 2 ijms-22-04317-f002:**
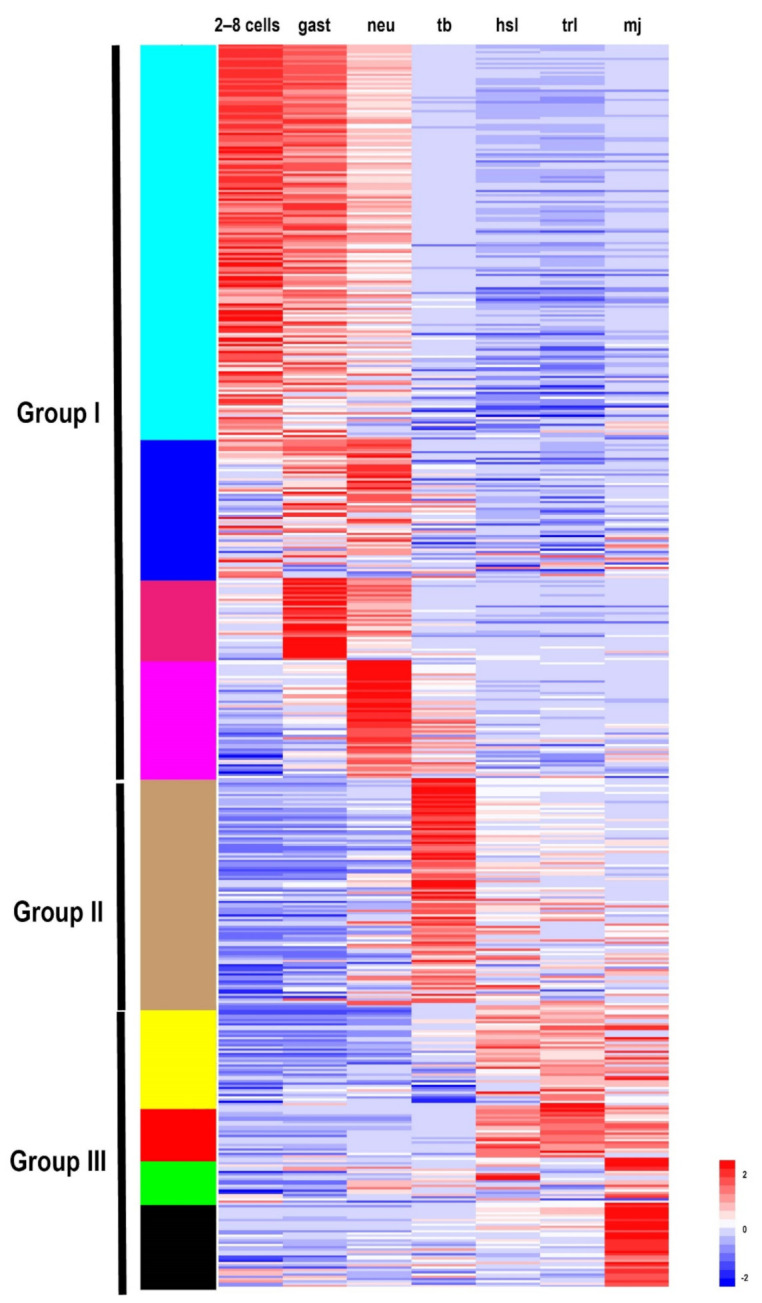
Expression patterns of TFs in *S. clava* genome—547 TFs in *S. clava* genome are shown in the heat map. They are classified into nine modules, including turquoise, blue, magenta, pink, brown, yellow, red, green, and black module through the WGCNA analysis. The turquoise, blue, magenta, and pink modules are classified into Group I, in which the TFs were highly expressed during 2–8 cells stage to the neural stage; the brown module is classified into Group II, in which TFs were highly expressed during the tailbud stage; the yellow, red, green, and black modules are classified into Group III, in which TFs were highly expressed from the hatched swimming larvae stage to the metamorphosis juvenile stage. The scale bar indicates the centered FPKM values. The abbreviation of different developmental stages are as follows—2-cell–8-cell embryos (2–8 cells), gastrula embryos (gast), neurula embryos (neu), tailbud-stage embryos (tb), hatched swimming larvae (hsl), tail-regressed larvae (trl), and metamorphic juveniles (mj).

**Figure 3 ijms-22-04317-f003:**
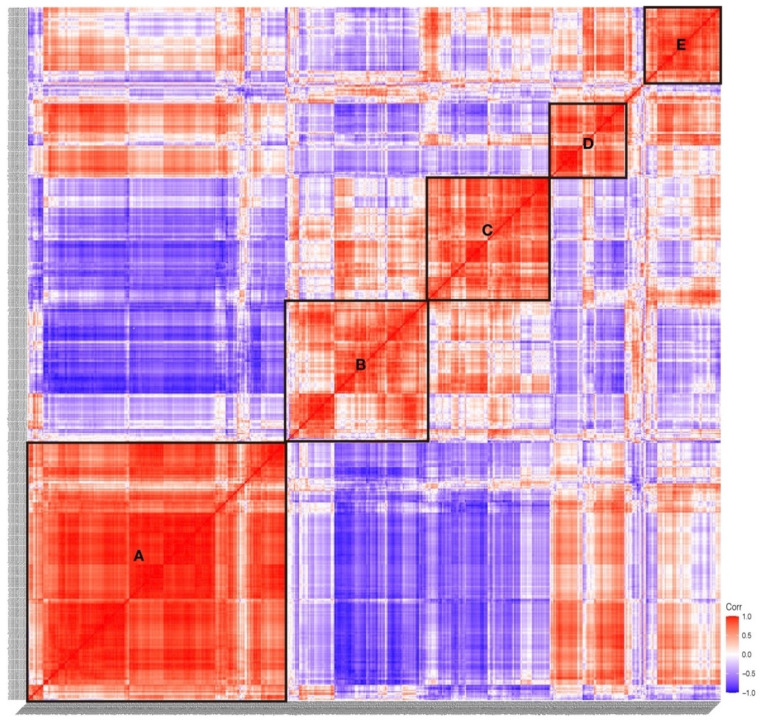
Expression correlation of TFs in *S. clava*. The correlation score showed between −1.0 (deep blue) to 1.0 (deep red). The red color indicated that these two TFs showed positive expression correlation, and the blue color showed negative expression correlation. Five significant correlation groups are indicated by box A to box E.

**Figure 4 ijms-22-04317-f004:**
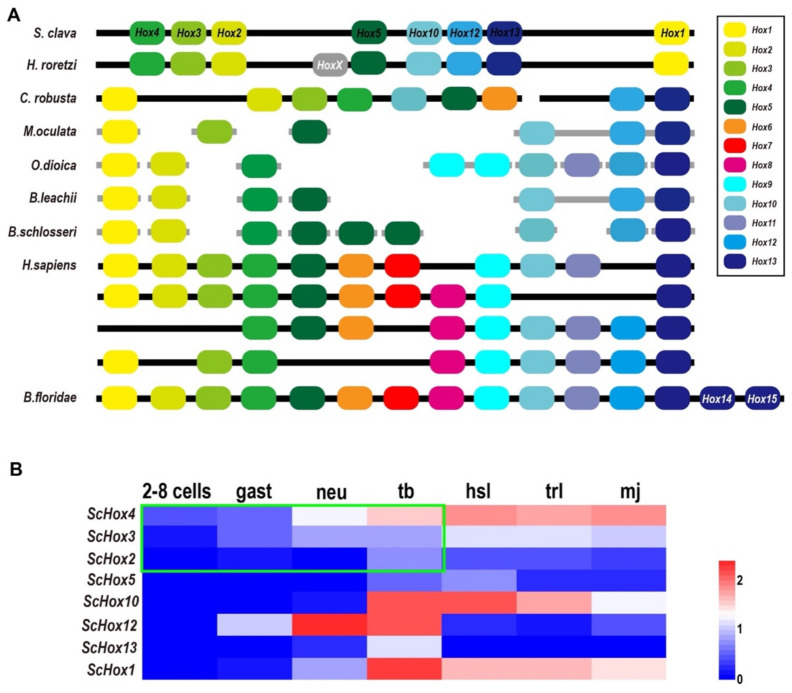
Comparison of *Hox* genes in different species and the expression patterns of *ScHox* genes. (**A**) Schema depicting *Hox* gene linkages retained in the genomes of seven tunicates, *H. sapiens* and *B. floridae*. Orthologous *Hox* genes are indicated by the same color of rounded rectangles and the legend is showed on the right. *HoxX* gene was an unclassified *Hox* gene in the *H. roretzi* genome. The spatial distribution of *Hox* genes in different species on the chromosome or scaffold are indicated by thick black line or thick gray line, respectively. (**B**) Heat map of the *ScHox* genes. Gene names are shown on the left. The scale bar indicates the FPKM values, which are dealt with log_10_(FPKM + 1) but not centered. The green box shows a subcluster-level temporal co-linearity (STC) expression pattern among the *ScHox2*, *ScHox3*, and *ScHox4* cluster. Shorthand of different developmental stage, which is showed above the heat map, is mentioned above.

**Figure 5 ijms-22-04317-f005:**
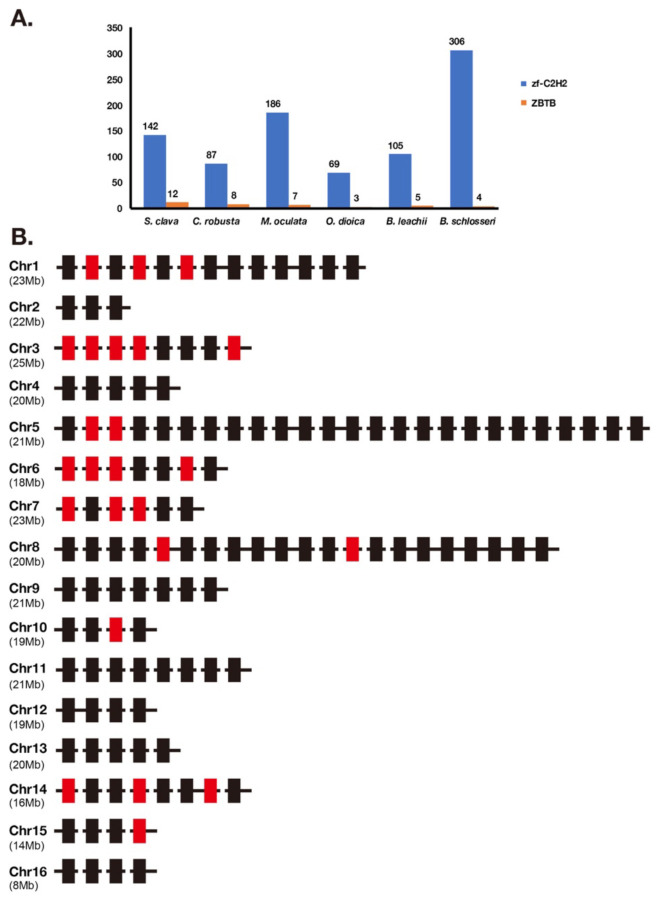
Classification of *zf-C2H2* genes. (**A**) Statistics of *zf-C2H2* genes and *ZBTB* genes in zf-C2H2 family in *S. clava*, *C. robusta*, *M. oculata*, *O. dioica*, *B. leachii*, and *B. schlosseri*. The blue volume indicates the number of *zf-C2H2* genes, and the orange volume indicates the number of *ZBTB* genes. (**B**) The distribution of *zf-C2H2* genes on chromosome. The rectangles indicated the *zf-C2H2* genes and the colors indicate the expression levels of each *zf-C2H2* genes, according to the WGCNA analysis (mentioned in Figure 2, the black and red rectangles indicate the high expression of the *zf-C2H2* genes during 2 cells to neu stages, and during the tbl to mj stages, respectively). The chromosomes are labeled on the left. The black horizontal line indicates scaffold.

**Figure 6 ijms-22-04317-f006:**
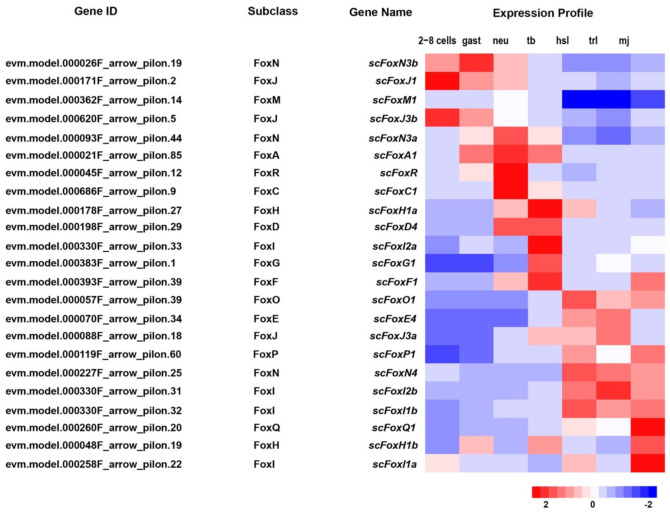
Identification and expression of *ScFox* genes in *S. clava*. The Gene IDs, subclasses, Gene names of 23 *scfox* genes are listed (left side). The expression profiles for each gene through different stages (2–8 cells, gast, neu, tb, hsl, trl, and mj stage) are shown on the right side. The scale bar indicates the centered FPKM values.

**Figure 7 ijms-22-04317-f007:**
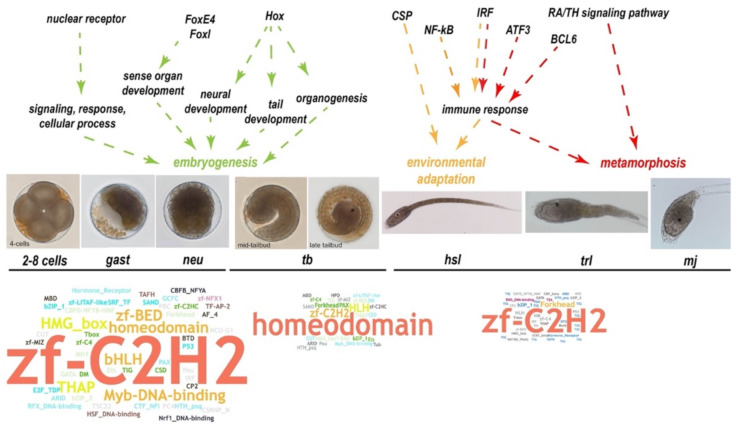
Summary of the TF expression and their potential roles in the embryogenesis and larval development of *S. clava*. Expression profiles indicates that TFs play essential roles during embryogenesis and larval development. We showed different stages of embryos at the bottom of the figure, including 2–8 cells stage, gast stage, neu stage, mid-tailbud stage, late-tailbud stage, hsl stage, trl stage, and mj stage. We summarized the predicted function of TFs genes into three parts and showed it in different colors, including embryogenesis (green), environmental adaptation (yellow), and metamorphosis (red). The word cloud pictures show the family of the expressed TF genes at different stages. The font size indicates the number of expressed TF genes in the TF families.

**Table 1 ijms-22-04317-t001:** Number of TFs and TF families in *S. clava*, *C. robusta*, *M. oculata*, *O. dioica*, *B. leachii*, *B. schlosseri*, *H. sapiens*, *B. floridae*, and *C. elegans*.

TF Superfamily	TF Family		*S. clava*	*C. robusta*	*M. oculata*	*O. dioica*	*B. leachii*	*B. schlosseri*	*H. sapiens*	*B. floridae*	*C. elegans*
		AF-4	1	1	1	1	1	1	8	1	0
		ARID	4	5	7	4	4	2	15	4	5
		BTD	2	2	2	2	2	3	6	2	1
	bZIP	bZIP_1	13	16	12	21	15	7	37	14	11
		bZIP_2	6	7	7	8	6	2	12	8	17
		CBF_beta	1	1	1	1	1	1	1	1	1
	NF-Y	CBFB_NFYA	1	1	0	1	1	0	1	1	2
		CBFD_NFYB_HMF	8	9	8	13	8	4	45	16	9
		CG-1	0	1	1	0	0	0	2	1	1
		CP2	2	3	2	2	2	3	8	2	1
		CSD	9	5	4	7	4	4	17	2	5
		CSRNP_N	1	1	0	2	1	2	3	1	1
	MH1	CTF_NFI	2	1	0	0	1	0	4	1	0
		MH1	4	5	7	11	4	5	11	4	7
		E2F_TDP	2	4	4	3	3	2	20	4	5
	ETS	Ets	12	15	12	11	14	12	29	12	10
		ETS_PEA3_N	0	0	0	0	0	0	4	1	0
		Forkhead box	23	25	24	26	27	27	55	28	16
		GCFC	2	2	2	1	2	2	3	1	2
		GCM	0	0	0	0	0	0	2	1	0
		GTF2I	0	1	0	0	0	0	15	0	0
	bHLH	bHLH	41	39	36	22	37	20	102	73	38
		Myc_N	0	1	1	0	0	1	4	1	0
		SIM_C	0	0	0	0	0	0	2	0	0
		HMG_box	20	20	14	19	18	21	102	29	16
	Homeobox	Homeodomain	73	76	74	70	74	45	264	108	85
		CUT	2	1	3	3	3	2	8	3	7
		PBC	1	1	1	1	1	5	11	1	2
		Pou	2	3	3	5	3	2	17	6	3
		HPD	2	2	1	1	1	1	3	0	1
		HSF_DNA-binding	1	1	0	4	1	2	9	5	1
		HTH_psq	6	2	3	0	9	0	2	5	1
	IRF	IRF	5	5	6	2	8	5	3	4	0
		IRF-3	1	3	2	0	4	9	9	4	0
		LAG1-DNAbinding	0	0	0	0	0	1	0	0	0
		LRRFIP	0	1	3	1	1	1	5	1	1
		MBD	3	2	4	0	3	2	7	3	2
		Myb_DNA-binding	12	13	11	7	13	11	25	16	7
		NCU-G1	1	0	1	0	1	2	1	1	0
		NDT80_PhoG	1	1	3	0	1	2	2	1	2
		Nrf1_DNA-binding	1	1	1	0	1	1	1	3	0
		P53	3	2	1	1	3	5	3	2	1
		PAX	7	6	5	8	5	2	9	5	9
		PC4	1	1	0	1	1	1	3	2	1
		RFX_DNA_binding	2	3	3	1	3	2	10	5	1
		RHD_DNA_binding	3	2	3	2	4	5	13	2	0
		Runt	1	1	1	1	1	2	3	1	1
		SAND	3	2	1	2	1	2	7	4	4
		SRF-TF	3	2	2	3	2	1	5	3	2
	STAT	STAT_alpha	0	0	0	1	0	2	2	0	0
		STAT_binding	2	2	2	2	2	2	5	1	2
		STAT_int	0	1	0	0	0	0	0	2	0
		TAFH	2	1	1	1	1	1	6	3	1
		T-box	10	8	8	8	8	13	19	10	19
		TEA	1	1	1	2	1	1	4	1	1
		TF_AP-2	1	2	1	1	1	2	5	1	4
		TF_Otx	0	0	0	0	0	0	1	0	0
		TIG	11	7	6	12	16	29	18	15	4
		TSC22	2	1	1	0	2	1	4	1	5
		Tub	1	1	1	2	1	1	5	2	2
		Vert_HS_TF	0	0	0	0	0	0	3	0	0
zinc finger		THAP	21	8	27	0	40	49	9	7	3
		GATA	5	4	4	5	5	2	17	7	12
		DM	5	2	0	2	7	3	7	8	11
	Nuclear Receptor	Hormone_receptor	6	6	6	7	5	9	28	10	151
		Androgen_receptor	0	0	0	0	0	0	1	0	0
		Oest_receptor	0	0	0	0	0	0	1	0	0
		Prog_receptor	0	0	0	0	0	0	1	0	0
		GCR	0	0	0	0	0	0	1	0	0
		zf-C4	15	12	14	33	11	21	32	21	124
		zf-BED	17	2	6	0	8	9	3	0	9
		zf-C2H2	154	95	193	72	110	310	748	742	58
		zf-C2HC	5	3	2	2	4	5	8	2	1
		zf-LITAF-like	4	1	2	5	3	2	4	11	14
		zf-MIZ	2	2	3	2	1	0	6	2	2
		zf-NF-X1	1	1	2	0	1	1	1	1	1
Total TFs			553	456	557	425	522	693	1867	1240	703
Total TF families			60	64	57	51	61	60	74	64	55

**Table 2 ijms-22-04317-t002:** Number of TFs in the WGCNA modules.

TF Family	Turquoise	Blue	Magenta	Pink	Brown	Yellow	Red	Green	Black	Total
AF-4	1	0	0	0	0	0	0	0	0	1
ARID	1	1	1	0	1	0	0	0	0	4
BTD	1	0	0	0	0	1	0	0	0	2
bZIP_1	3	0	0	0	2	3	0	0	5	13
bZIP_2	3	2	0	0	0	0	0	0	1	6
CBF_beta	0	0	0	0	0	0	0	1	0	1
CBFB_NFYA	1	0	0	0	0	0	0	0	0	1
CBFD_NFYB_HMF	2	1	1	2	0	2	0	0	0	8
CP2	1	0	0	0	0	0	0	0	1	2
CSD	2	1	1	0	1	1	3	0	0	9
CSRNP_N	0	0	0	1	0	0	0	0	0	1
CTF_NFI	2	0	0	0	0	0	0	0	0	2
CUT	0	1	0	0	1	0	0	0	0	2
DM	1	1	2	0	1	0	0	0	0	5
E2F_TDP	1	0	0	1	0	0	0	0	0	2
Ets	3	0	0	3	2	3	0	1	0	12
Forkhead box	4	1	0	3	5	4	3	1	2	23
GATA	4	0	0	0	0	1	0	0	0	5
GCFC	2	0	0	0	0	0	0	0	0	2
HLH	10	7	0	3	13	2	3	0	3	41
HMG_box	9	2	1	2	2	0	0	2	2	20
Homeodomain	5	4	2	6	37	8	3	4	3	72
Hormone_recepter	2	0	0	0	0	2	1	0	1	6
HPD	0	0	0	0	1	0	0	0	1	2
HSF_DNA-binding	1	0	0	0	0	0	0	0	0	1
HTH_psq	0	2	1	0	1	0	0	2	0	6
IRF	2	0	0	0	0	0	0	0	3	5
IRF-3	0	0	0	0	0	1	0	0	0	1
MBD	1	0	0	0	1	1	0	0	0	3
MH1	3	0	0	1	0	0	0	0	0	4
Myb_DNA-binding	6	3	1	0	1	0	0	0	0	11
NCU-G1	1	0	0	0	0	0	0	0	0	1
NDT80_PhoG	0	0	0	0	0	1	0	0	0	1
Nrf1_DNA-binding	1	0	0	0	0	0	0	0	0	1
P53	2	0	0	0	0	1	0	0	0	3
PAX	2	0	0	1	3	0	1	0	0	7
PBC	1	0	0	0	0	0	0	0	0	0
PC4	0	0	1	0	0	0	0	0	0	1
Pou	1	0	0	0	1	0	0	0	0	2
RFX_DNA_binding	1	0	0	1	0	0	0	0	0	2
RHD_DNA_binding	0	0	0	0	0	1	0	0	2	3
Runt	0	0	0	0	0	0	0	0	1	1
SAND	1	1	0	0	1	0	0	0	0	3
SRF-TF	2	1	0	0	0	0	0	0	0	3
STAT_alpha	0	0	0	0	0	0	0	0	0	0
STAT_binding	0	0	0	0	0	1	0	0	1	2
TAFH	1	0	0	0	0	0	0	0	1	2
T-box	1	1	3	1	3	1	0	0	0	10
TEA	0	0	0	0	0	0	0	1	0	1
TF_AP-2	0	1	0	0	0	0	0	0	0	1
THAP	6	1	0	4	6	0	0	2	2	21
TIG	6	0	1	0	2	0	0	1	1	11
TSC22	2	0	0	0	0	0	0	0	0	2
Tub	0	0	0	0	1	0	0	0	0	1
zf-BED	3	2	0	7	2	2	0	1	0	17
zf-C2H2	67	24	20	12	7	5	7	3	5	150
zf-C2HC	2	2	0	0	1	0	0	0	0	5
zf-C4	0	1	1	3	2	3	2	1	2	15
zf-LITAF-like	1	1	0	1	1	0	0	0	0	4
zf-MIZ	1	0	0	0	1	0	0	0	0	2
zf-NF-X1	1	0	0	0	0	0	0	0	0	1
Total	174	61	36	52	100	44	23	20	37	

## Data Availability

The genome sequences of *S. clava* were deposited in NCBI under the BioProject number PRJNA523448. The transcriptome data of *S. clava* used for biological analysis were also deposited in the NCBI SRA database, with the accession numbers SRR8599814 to SRR8599834. The genome resources of *C. robusta*, *H. sapiens*, and *C. elegans* were downloaded from Ensembl (https://asia.ensembl.org/index.html), the genome resources of *O. dioica* and *B. floridae* were acquired from NCBI (https://www.ncbi.nlm.nih.gov/, accessed on 1 July 2020), and the genome resources of *M. oculata*, *B. leachii*, and *B. schlosseri* were acquired from ANISEED (https://www.aniseed.cnrs.fr/, accessed on 1 July 2020).

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
