# Peer review of "Genome-Wide Identification, Comparison, and Expression Analysis of Transcription Factors in Ascidian Styela clava"

_ijms, 2021, doi:10.3390/ijms22094317_

Round 1

Reviewer 1 Report

The authors of this manuscript present an analysis of the transcription factor compliment in Styela clava. Based on genome and transcriptome analyses, they report 553 transcription factor genes in 60 different families (from 17,428 total identified genes).

The data presented primarily consisted of gen and gene family counts, with little analysis or interpretation of the significance of the findings. For example, in figure 1 the authors report that 51 gene families are shared among representative metazoans and 51 gene families are shared among 5 species of tunicates. It is not clear that these families are the same between these two groups; looking at the supplemental file provides insight that while similar, these are not identical families. It would have been nice to highlight the shared families and differences in this table.

The authors also state that there are more ZnFn genes identified in Styela vs other tunicates doesn’t have much context – is this to be expected based on differences in genome size or on gene number?  Are these differences proportional to differences in genome size/gene number? Do these expanded genes have known functions that could explain why there are more ZnFn genes in Steyla vs the other tunicates?

Figure 2 presents time-course expression data for various TF genes – how well does this correlate from findings from other ascidians or other embryos (i.e. the sea urchin embryo)? Are the same patterns of TF expression during embryogenesis seen in these different species? Why or why not? Is there a pattern of TFs that could explain any differences?

It is not clear what figure 3 represents and the author should more clearly describe what they are depicting. In general, the information presented in figures 2 and 3 would be difficult to use in any other analyses by other researchers because no gene sequences/gene models are presented. It seems that this information should be available as the genome has been sequenced and the gene models assembled for these types of analyses.

Figure 4 shows the location of the Hox gene clusters in various organisms. The authors show that the cluster organization is similar in Styela and Halocynthia, but fail to mention that these two species are more closely related to each other compared to Ciona. There is no analysis/interpretation of the organizational arrangement of the Hox genes. A time course of expression is also shown, how does this compare to the known patterns of Hox expression in other tunicates?  Are they similar/different?  What is the significance?

In figure 5 it is not clear what the significance is of ZnFn gene locations along the chromosomes.

Figure 6 reports a time course of Fox genes, again what is the significance compared to other species?

Reviewer 2 Report

In this manuscript, Zhang and colleagues describe the repertoire of Transcription factors (TFs) in the ascidian Styela clava and present a preliminary analysis of their evolution and expression. TF repertoires such as the one presented here are very useful community resources. This referee supports the publication of this study in the Int. J. Mol. Sci., provided the authors address the following methodological and redactional concerns.

Content and scope of the study:

  1. Manuscript focus:

It is unclear to this referee what led to the authors to highlight The Fox family in their analyses. The authors should explain this specific choice as opposed to choosing Sox, bHLH, bZip or the non-hox HD family. As no significant message comes out of the Fox analysis, it could be suppressed (text, figures, supp figures). and replaced by a deeper analysis of the evolution and expression (see 2) of the whole TF repertoire, the Hox and the BTB families.

  1. Introduction:

    1. Line 39, the “nucleotide sequences” of TFs tend to be very poorly conserved. Even at the aa level, the only conserved domain is frequently the DNABD.

    2. The authors should be more precise line 59 when they write that “Styela presents strong environmental adaptation”. Can they at least provide a reference in support of this claim?

  2. Results:

    1. Line 91. I do not see a contraction of THAP genes in Hsap, but rather a specific expansion in stolidobranch ascidians.

    2. Line 95: Table 1 shows that there is not LAG1 in ascidians. Searching table S2 with “Lag1”, “lin-12” or “glp” also returns no result. It does not seem that Lag1 is present in tunicates

    3. Lines 107-108: please tone down. This is at best suggestive of adaptative changes.

    4. Lines 131 to 141: I am surprised that so few Styela TFs are expressed throughout embryogenesis. Please confirm.

    5. Figure 3: this is a nice analysis, but I am not sure that the order of the genes along the axes is the best one. Bringing together the genes in Blocks A, D and E would much more efficiently show the anticorrelation between the early (A, D, E) and late (B, C) TF programs.

    6. Hox gene analysis: the authors could deepen a bit the interpretation of their results and comment on the unexpected order of the genes, and on the lack of temporal collinearity.

    7. Figure 5: indicating the length of the chromosomes would strengthen the claim of an enrichment on Chr5.

  3. Discussion:

    1. Line 256: are all ascidians invasive? Or are the authors specifically referring to Styela?

    2. Line 261: I do not see any ascidian amplification of RHD_DNA-bind in Table 1. The CSD amplification is mild. Please tome down or remove.

    3. Lines 272-273: the best example of such as phenomenon is the maternal ETS1/2 TF previously studied by the Nishida, Lemaire and Satou groups. This should be mentioned, and the adequate studies referred to.

    4. Lines 280-281: the involvement of immune response genes in metamorphosis has already been proposed and these earlier studies should be cited.

    5. Lines 295-296: I see no evidence for any type of Hox temporal collinearity in the data shown.

    6. Line 308: this is more than expected as the teleosts have undergone 3 rounds of whole genome duplication… Please remove.

    7. Line 316: gene expression is never sufficient to “infer” the function of a gene... At best is provides a possible clue.

Methodological concerns:

Some methodological explanations are missing, precluding reproducibility of the study, and some methodological choices may weaken the interpretations that can be drawn.

  1. RNA-seq analysis: The authors consider that any gene with an FPKM value >1 is expressed. This is very close to background level and the WGCNA software recommends considering genes to be expressed if their FPKM value is >10. The authors should justify their choice of a lower value and show that using a FPKM >10 threshold would not alter their conclusions.

  2. RNA-seq analysis: As the raw RNA-seq data reanalyzed here have already been published in a previous article by the Dong lab (https://doi.org/10.1111/1755-0998.13209), there is no need to describe the collection, sequencing and processing of the data again in this article. Please suppress the corresponding methods lines (336-343).

  3. GO enrichment: the authors should describe how the GO analysis shown on Figure S3 was carried out: software, parameters used and most importantly what was the reference gene set the maternal TFs were compared to when computing enrichment? Is it the whole genome complement of coding genes? Or all TFs? Also is the result shown specific for maternal TFs or would a similar analysis with TFs expressed zygotically only return the same enriched GO terms? Can the authors also explain what the Rich Factor is (with corresponding reference)?

  4. Traceability of the results: In Table S2, the authors may want to add the gene names indicated in some of the main text figures (e.g. scHox4…). Table S1, the pfam domains (and their Regulome or AnimalTFDB origin) used to identify each family should be indicated.

  5. Traceability of the results: The ENSEMBL version used for download should be indicated. It is also good practice to cite the relevant article for ENSEMBL, NCBI, corrplot and ANISEED, and not just their URL.

  6. Phylogeny analyses: The phylogenetic tree shown in Figure S2 only includes 3 species. Phylogenies based on such small number of species can be unreliable in cases of gene losses. The authors should rerun their analysis using at least 2 non mammalian vertebrates in addition to Hsap and the 5 tunicate species considered. This would very significantly improve the quality of the resource.

  7. Phylogeny analyses: some gene orthology assignments are not supported by the phylogenetic trees presented: e.g. scHox5 and scHox10 (see phylogeny Figure S2). If the authors keep the Fox section, it would be useful to provide a phylogenetic tree to support their naming.

  8.  

Redactional issues:

The manuscript could be substantially improved by correcting some mistakes in figures and legends. It should also be read and corrected by a native English speaker (eg “high” and “highly” are not the same, neither are “essential” and “essentially”, etc..).

  1. Figure mistakes: In Figures 2 and 6, the Log10(FPKM +1) values are shown to run from -2 and +2 or -1 to +1. By construction, Log10(FPKM +1) has to be positive. The authors should correct their legend to indicate what the scale shown really represents (This referee’s guess is that the values shown have additionally been centered and normalized, but this may be wrong).

  2. Figure mistakes: Figure 1 legend: Venn not Veen diagram.

It would have been nice to point to a genomic browser to visualize the data

Reviewer 3 Report

The manuscript “Genome-wide identification, comparison and expression analysis of transcription factors in ascidian Styela clava” is a good scientific soundness paper, written in detail and which provides useful information on transcription factors from an ascidian species. Some small improvements might be added. Namely, in the introduction section, the evolutionary meaning of different TFs with specific examples should be included. In the discussion section, the relation of TFs with the fitness of Tunicates could be explored. Some parts of the manuscript need further attention, for example, a connection between statements is missing in lines from 257 to 264.  

Detailed:

Line 256 on page 9 - should be re-written as “As an invasive species, ascidians showed…” is not a correct statement. Not all ascidians are invasive, S. clava may have an invasive behaviour but it depends on the geographic region and so on.

Line 258 on page 9 – what do you mean by “TFs were expanded”. Please clarify the statement. 

Round 2

Reviewer 1 Report

The authors have tried to address many of my concerns, however I still believe that there are missed opportunities to improve the significance of the manuscript.